# Particularities of Catheter Ablation in Women with Atrial Fibrillation and Associated Ischemic Heart Disease

**DOI:** 10.3390/jcm11195568

**Published:** 2022-09-22

**Authors:** Diana Andrada Irimie, Adela Viviana Sitar-Tăut, Bogdan Caloian, Florina Frîngu, Gabriel Cismaru, Radu Roşu, Mihai Puiu, Ioan Alexandru Minciună, Gelu Simu, Dumitru Zdrenghea, Dana Pop

**Affiliations:** 1Internal Medicine Departament, Rehabilitation Cardiology Discipline, “Iuliu Hațieganu” University of Medicine and Pharmacy, 400337 Cluj-Napoca, Romania; 2Cardiology Department, Clinical Rehabilitation Hospital, 400347 Cluj-Napoca, Romania

**Keywords:** catheter ablation, atrial fibrillation, ischemic heart disease, female sex

## Abstract

Background: Atrial fibrillation is more common in men, but in the presence of ischemic heart disease, this arrhythmia is more frequent in women. However, like in coronary heart disease, women with atrial fibrillation are suboptimally treated. Methods: To identify particularities of ablation, in women with atrial fibrillation and ischemic heart disease. Results: 29 women and 26 men, with documented ischemic heart disease and atrial fibrillation, who underwent catheter ablation, were admitted in the study. No significant differences were registered regarding the heart rate control treatment. Electrical cardioversion was significantly higher in men, while pharmacological cardioversion was predominantly recommended in women. The ablation was performed later in women, after 2.55 ± 1.84 years versus 1.80 ± 1.05 in men (*p* = 0.05). The time elapsed until the ablation was performed was statistically correlated with atypical symptomatology and with the number of antiarrhythmics used prior to the ablation. There were no significant differences for the relapse of atrial fibrillation at 3 months. Quality of life at 3 months after ablation was increased in both groups. Conclusion: Catheter ablation is performed much later in women, and the causes responsible for this delay would be more atypical symptoms and a greater number of antiarrhythmics tried before the ablation.

## 1. Introduction

The relationship between ischemic heart disease and atrial fibrillation is a much studied one, the interrelationship between these two pathological entities being a bidirectional one. Myocardial ischemia creates favorable conditions for the appearance of atrial fibrillation through electrical instability secondary to the complex interaction between ionic, metabolic, and neurohormonal changes in cardiac cells, thus increasing the proarrhythmic risk, and ischemia is also a main pillar of the initiation and progression of atrial fibrosis [1]. On the other hand, atrial fibrillation itself is a promoter of fibrosis and atrial structural remodeling, changes with significant consequences also on the coronary circulation. Therefore, a vicious circle is created between these two conditions: ischemia generates myocardial fibrosis and vice versa [2].

Atrial fibrillation is more common in men, but it seems that when we talk about the incidence and prevalence of atrial fibrillation in patients with coronary artery disease, we are talking about a more frequent association of this arrhythmia in female patients. At the same time, the REGARDS study demonstrates a higher risk of myocardial infarction in women with atrial fibrillation, but also a higher risk of developing atrial fibrillation in women with myocardial infarction [3].

Unlike men, in women, this rhythm disorder appears at a much older age, after menopause is established, and hormonal changes secondary to menopause lead to an increase in inflammatory markers with structural and electrophysiological implications. Thus, in these women there is a much higher level of inflammation compared to men, because to the degree of inflammation conferred by the atherosclerotic disease itself, there is also the presence of hormonal influences, with the loss of cardiovascular protection conferred until that moment by the level of estrogen. There is also a greater degree of myocardial fibrosis [4].

According to the ESC guidelines for atrial fibrillation, treatment recommendations do not differ according to the patient’s gender [5]. However, in clinical practice there are differences regarding drug treatment, and here we refer to the medication used for rhythm control, rate control, to prevent thromboembolic events, but also regarding interventional treatment or electrical cardioversion. The only mentions regarding the patient’s gender are related to the higher risk of recurrence of atrial fibrillation after cardioversion in women [5].

In this context, the present study aims to identify therapeutic particularities, especially those related to the interventional treatment of atrial fibrillation, in women with ischemic heart disease.

## 2. Materials and Methods

We included in the study a total of 55 patients with documented ischemic coronary heart disease and atrial fibrillation, admitted to the Cardiology Department of the Clinical Rehabilitation Hospital, during 2019–2020, in order to perform the atrial fibrillation catheter ablation procedure. From the total of 55 patients, 29 were women and the remaining 26 were men, so the patients were divided into two groups, depending on gender. Ischemic heart disease was defined according to the Guideline of Chronic Coronary Syndromes of the European Society of Cardiology published in 2019, with patients presenting both ischemic heart disease through microvascular damage and atherosclerotic obstructive coronary disease.

Before performing the catheter ablation procedure, the demographic and biological parameters and the associated comorbidities were evaluated and a thorough anamnesis was performed regarding the history of the rhythm disorder, its clinical manifestations, and the history regarding the therapeutic strategies carried out up to the moment of ablation. According to the recommendations of the Atrial Fibrillation Guideline, of the European Society of Cardiology, all patients were pre-procedurally evaluated by transthoracic, transesophageal ultrasound, and cardiac CT angiogram. At the transthoracic ultrasound, the size of the left atrium and the left ventricle was analyzed, LVEF% (left ventricle ejection fraction) was calculated using the Simpson biplane method, diastolic dysfunction, the presence of mitral insufficiency, and pulmonary arterial hypertension were evaluated. Transesophageal ultrasound was performed to exclude the presence of an intracardiac thrombus, and LAA velocity (left atrial appendage), LAA size, and the integrity of the interatrial septum were analyzed to guide the intraprocedural transseptal puncture.

For the catheter ablation procedure, the isolation of the pulmonary veins was carried out with the help of CARTO 3D-Biosense Webster Inc. mapping systems. and Ensite NavX-St. Jude Medical Inc. Of the total ablated patients, 89.1% (49 patients) performed radiofrequency ablation and the remaining 10.9% (6 patients) performed cryoablation.

Intraprocedural parameters, such as the need for intraprocedural electrical cardioversion, were also evaluated. Later, after 3 months, the effectiveness of the ablative treatment was evaluated by analyzing the presence of the recurrence of atrial fibrillation and the quality of life of the patients, as well as the factors associated with the risk of the recurrence of atrial fibrillation episodes. The patients’ quality of life was evaluated based on a questionnaire, which was completed by each patient separately and which included questions about the improvement in symptoms, about the presence of complications and the recovery period, about the experience of the intervention, about its improvement or absence after the procedure, but also about the reasons responsible for the lack of improvement in the quality of life.

The statistical packages MedCalc 10.3.0.0 (MedCalc Software, Ostend, Belgium) and SPSS for Windows 16.0 (IBM Corporation, Armonk, New York, NY, USA) were used for statistical data processing. In the case of numerical variables, after testing the normality of the distribution (using the Kolmogorov–Smirnov and D’Agostino–Pearson tests), averages, standard deviations, and medians were determined. Qualitative data were reported in the form of numbers and percentages. Independent sample t-test, Mann–Whitney, ANOVA (analysis of variance), and Kruskal–Wallis, respectively χ^2^, tests were used to identify differences between variables or groups (depending on the type of variable). The relationships between the variables were highlighted by calculating the Spearman or Pearson correlation coefficients, respectively, the univariate or multivariate regression. A value of *p* < 0.05 was considered statistically significant.

## 3. Results

The main characteristics of the patients included in the study are summarized in Table 1.

All patients included in the study underwent transthoracic ultrasound, transesophageal ultrasound and also heart CT angiography. Table 2 shows the ultrasound and imaging parameters.

As can be seen in this table, analyzing the cardiovascular risk factors, smoking, alcohol consumption, and hyperuricemia have higher values in men. Microvascular angina predominates in women, and macrovascular angina in men. 

The main aim of our study was to highlight the differences regarding the treatment of atrial fibrillation between the two sexes, knowing that women are undertreated, both from the point of view of ischemic coronary disease, and from the point of view of disorders of rhythm. The treatment of atrial fibrillation involves three objectives: prevention of thromboembolic events, heart rate control, and heart rhythm control. Various aspects related to the prevention of thromboembolic events are detailed in Table 3.

One of the most feared complications of atrial fibrillation is represented by the occurrence of thromboembolic stroke. The CHA2DS2-VASC score had a median value of 3, this value being a clear indication for oral anticoagulant treatment, regardless of the patient’s gender. In our study, we analyzed the incidence of stroke, both in the general population and according to the patient’s gender. We also evaluated whether the occurrence of thromboembolic stroke correlates with the type of atrial fibrillation. Figure 1 shows the correlations between the type of atrial fibrillation and the presence of ischemic stroke by thromboembolic mechanism. The type of atrial fibrillation did not correlate with the presence of stroke, but compared by gender, the prevalence of stroke was recorded in a higher percentage in men than in women, 26.9% (7 men) versus 13.8% (4 women), but without the difference between the two sexes being statistically significant (*p*-0.087).

For heart rate control, the most commonly used drug class is represented by beta-blockers. An alternative to the administration of beta-blockers is the administration of non-dihydropyridine calcium-channel blockers. When the heart rate is insufficiently controlled with beta-blockers, digoxin is added to the therapeutic plan. In Table 4, we present the treatment for heart rate control compared between the two sexes.

However, when we compared the type of beta-blocker administered according to the patient’s gender, we observed that the differences were statistically significant, with bisoprolol being administered in a significantly higher percentage in men, and metoprol being predominantly prescribed among women (*p* = 0.05). 

As we said before, the treatment of atrial fibrillation has objectives, and the third objective, and perhaps the most important of all, is represented by heart rhythm control. Heart rate control can be achieved pharmacologically, through electrical conversion and through the ablation procedure. All patients in our study underwent the ablation procedure. Figure 2 shows the distribution of the main types of antiarrhythmics compared between the two sexes.

Before the catheter ablation, pharmacological cardioversion was achieved in a higher percentage in women (44.82%), compared to the male sex where it was achieved in 34.61%, but without statistical difference. Instead, the electrical cardioversion was achieved in a significantly higher percentage in men compared to women (57.69% versus 27.58%, *p* = 0.047).

Regarding ablation as a treatment for heart rate control, it appears to be performed much later in women. In our analysis, the interventional treatment was performed on average after 2.55 ± 1.84 years in women, statistically significantly later than in men, where the average time to the first catheter ablation procedure was only 1.80 ± 1.05 years (*p* = 0.005). Differences were also recorded regarding the indication for ablation, namely, the indication for paroxysmal but also persistent atrial fibrillation was more frequent in males, compared to females, where the indication for interventional treatment was predominantly in the case of recurrent paroxysmal atrial fibrillation. Table 5 details aspects of heart rate control treatment.

In women, both the symptoms of ischemic coronary disease and those related to atrial fibrillation are often atypical. In our study, it was demonstrated that the main clinical manifestations of atrial fibrillation in women were represented by decreased exercise tolerance and dyspnea, which were recorded in a significantly higher percentage in them, compared to male patients. We correlated the symptomatology of all the patients included in the study and the number of years needed until the catheter ablation procedure was performed and we noticed that there was a direct correlation in the case of decreased exercise tolerance. No statistical correlations were recorded between the symptomatology represented by dyspnea or angina pectoris and the time until the ablation was performed. When we analyzed these correlations according to the patient’s sex, precisely to elucidate the reason for the later performance of ablation in women, we did not obtain any statistical correlation. In Figure 3, Figure 4 and Figure 5, the correlations between the symptoms of the patients and the number of years until the ablation is performed, both in the general population included in the study and according to the sex of the patients, are illustrated.

We also obtained a direct correlation between the number of antiarrhythmics used and the time elapsed until the ablation was performed, but only globally (Rho = 0.294, *p* = 0.00303, Figure 6A), and not depending on the patient’s gender (female: Rho = 0.191, *p* = 0.3112; male gender: Rho = 0.209, *p* = 0.2958, Figure 6B)).

Table 6 provides details on the risk factors assessed for the risk of recurrence of atrial fibrillation 3 months after catheter ablation.

Additionally, at 3 months, the patients were also evaluated regarding the after-procedural quality of life, whether it improved or not and whether they considered the interventional treatment of atrial fibrillation beneficial. Figure 7 shows the relationship between patients’ gender, quality of life, and the presence of atrial fibrillation recurrence 3 months after ablation.

## 4. Discussion

Following the assessment of the main characteristics of the patients included in the study, we managed to outline a gender-differentiated profile of patients with atrial fibrillation and associated ischemic coronary artery disease. No differences were observed regarding the mean age of the patients, but male patients were more frequent smokers and alcohol drinkers compared to female patients. Smoking is one of the risk factors with an important role in the development of atrial fibrillation and ischemic heart disease, but also a factor that decreases the effectiveness of both the treatment for controlling heart rate and heart rhythm [5]. Alcohol consumption is recorded in a higher proportion in the male sex, being an important risk factor for atrial fibrillation [5]. HDL-cholesterol was also lower in men. The relationship between hypo-HDL-cholesterol and atrial fibrillation is a bidirectional one, namely, the low level of HDL-cholesterol increases the risk of rhythm disorders, especially in patients with ischemic heart disease, but at the same time, atrial fibrillation itself leads to lowering of the level of HDL-cholesterol [6,7]. Hyperuricemia is a known independent risk factor of atrial fibrillation, being associated with an increased incidence of paroxysmal and persistent atrial fibrillation. A study published in 2018 demonstrated a higher incidence of atrial fibrillation in women with hyperuricemia [8]. The results of our study revealed a higher level of uric acid in men, which could be explained by the presence of obstructive atherosclerotic ischemic coronary disease in the patients we evaluated.

Women predominantly presented the microvascular angina, while obstructive atherosclerotic coronary disease is much more common in men, and in the latter a history of myocardial infarction is recorded in a significantly greater proportion than in women. There is a direct interconnection between ischemic heart disease through microvascular damage and atrial fibrillation. Coronary microvascular dysfunction is responsible for neurohormonal, but also sympathetic activation and myocardial fibrosis, all of which have a major role in the pathophysiological process of atrial fibrillation. The association of the two conditions accelerates the evolution towards heart failure [9].

In the present study, it was demonstrated that the main clinical manifestations of atrial fibrillation in women were represented by decreased exercise tolerance and dyspnea. These less specific and obvious manifestations can delay the presentation to the doctor, with possible consequences in the later diagnosis of this arrhythmia and the delay in the interventional treatment of atrial fibrillation.

We underline the fact that women subjected to the catheter ablation procedure presented a significantly higher percentage of recurrent paroxysmal atrial fibrillation, while men had more frequent paroxysmal and persistent atrial fibrillation. In men, this procedure is recommended much faster, which explains the higher percentage of paroxysmal fibrillation in them. At the same time, in women, the time elapsed from the onset of atrial fibrillation to the interventional treatment is longer, approximately 6.5 years in women compared to 5 years in men. This result confirms the data from the literature, according to which women are subjected to interventional treatment after a longer history of atrial fibrillation [10].

From an echocardiographic point of view, men presented more important left atrial dilatation compared to women. It is well known that normally in men the dimensions of the cardiac cavities are larger than in women [11]. Transesophageal ultrasound also revealed larger dimensions of the LAA in men than in women. We observed a significantly higher LAA velocity value in male patients. This result confirms two hypotheses about the differences between women and men regarding atrial fibrillation. The first is represented by the fact that women present a higher embolic risk, explained on the one hand by these lower blood-flow velocities at the LAA level and therefore a higher risk of spontaneous contrast or intraauricular thrombus formation. The second validated hypothesis is the one according to which men have both a lower thromboembolic risk than women, as well as higher chances of successful conversions and maintaining sinus rhythm compared to women, explained by the presence of higher blood flow velocities in these patients.

During the cardiac CT angiogram evaluation, a higher prevalence of pulmonary vein anomalies was observed in the female sex, specifically the presence of a common left-sided trunk. Given the small number of patients included in the study, it would be difficult to say that this anatomical variation of the pulmonary veins is specific to the female sex, but it could support the hypothesis that anatomical variations of the pulmonary veins could predispose to the occurrence of atrial fibrillation [12].

In our study, oral anticoagulant treatment was prescribed to women in a slightly lower percentage than to men. They received a higher percentage of VKA compared to NOAC. Bleeding risk assessed by means of the HAS-BLED score was similar for both sexes.

Analyzing the entire group, the type of atrial fibrillation in patients undergoing the ablation procedure did not correlate with the presence of ischemic stroke. In women, ischemic stroke was present only in patients with paroxysmal and recurrent atrial fibrillation, while in men, the prevalence of stroke was similar in the case of paroxysmal, recurrent, and persistent atrial fibrillation.

Preprocedural pharmacological treatment was not different between the two sexes (beta-blockers, calcium-channel blockers, and digoxin). Differences between the two sexes were only recorded when we evaluated the type of beta-blocker used, Bisoprolol being administered more frequently in men, while Metoprolol more frequently in women.

Before the catheter ablation in women, pharmacological cardioversion was attempted in a higher percentage compared to men, but without this difference being statistically significant. In contrast to pharmacological cardioversion, electrical cardioversion was achieved in a significantly higher percentage in men. This fact is described and demonstrated by other studies in the literature, and the explanations underlying this phenomenon were represented by the lower incidence and prevalence of atrial fibrillation in women, being less symptomatic or presenting more atypical symptoms and at the same time more in old age, which implies a higher risk of complications [13]. In the RACE study, it was proven that there is a worse prognosis in women in whom heart rhythm control was achieved, the results of this study being able to influence the reduction of the percentage of women proposed for electrical cardioversion [14]. At the same time, there are other data that demonstrated a higher risk of stroke in women with an increased CHA2DS2-VASc score who underwent electrical cardioversion [13]. However, despite these hypotheses, it is still difficult to say that the choice to achieve the conversion to sinus rhythm by administering the external electric shock is directly influenced by the patient’s gender.

In our study, catheter ablation was performed much later in women than in men. We cannot say that for the male gender the time of approximately 2 years is satisfactory. These delays are mainly influenced by socio-economic reasons such as the high costs of the procedure and long waiting lists, the longer duration of the procedure, which reduces the possibility of performing a large number of ablations per day. The differences between the two sexes do not stop only at the short time from establishing the diagnosis of fibrillation to the actual performance of the interventional treatment but are also observed at the indication of the catheter ablation procedure. More precisely, in women, catheter ablation is performed predominantly for recurrent atrial fibrillation, while in men, predominantly for paroxysmal and persistent atrial fibrillation.

Regarding the type of ablation used, radiofrequency catheter ablation predominated, compared to cryoablation, with no differences between the two sexes regarding the choice of technique used. Given the fact that there is a predominance of patients with recurrent and persistent atrial fibrillation addressed to interventional treatment, the use of radiofrequency ablation would be required due to the need for a more extensive ablation, it is not limited only to the isolation of the pulmonary veins, but also requires the ablation of other areas, such as that of the mitral ring, the Marshall vein, the posterior wall of the left atrium, etc.

Since in women the symptoms of atrial fibrillation can be less specific, we evaluated whether this could represent a cause of delay in ablation. Decreased exercise tolerance as a symptom was a parameter that was associated with longer time from diagnosis to catheter ablation, but this was only observed when we analyzed the entire population included in the study, without any correlation depending on the gender of the patient. An explanation for the absence of correlation based on the patient’s sex could be the small number of patients included in each group presenting this symptom. Angina, dyspnea, and the presence of palpitations were not associated with the delay in performing the ablation procedure.

The use of multiple classes of antiarrhythmic drugs in our study was not associated with time to ablation when we assessed this correlation by patient gender, but it was found to be a cause when we assessed the overall population of the study. Men tried more antiarrhythmics than women before the ablation procedure, even if the differences were insignificant from this point of view between the two sexes.

In the Atrial Fibrillation Guideline of the European Society of Cardiology published in 2020, several predictive risk scores for the recurrence of atrial fibrillation after ablation are presented in an additional appendix, and some of them such as CHA2DS2-VASc, DR-FLASH, MB- LATER, ATLAS, or CAAP-AF also considered the female gender as a risk factor for recurrence [5]. In our study, female gender did not prove to be a risk factor for the recurrence of atrial fibrillation at 3 months after ablation. In addition to the classic risk factors, we tried to evaluate if there are other new risk factors that can influence the risk of recurrence, such as the type of atrial fibrillation, the inflammation represented by the level of uric acid and the level of LDL-cholesterol, the velocity of blood flow at the level of the LAA, the type of ischemic heart disease, or thyroid pathology. Among them, the velocity of the blood flow from the LAA was found to be associated with the risk of relapse 3 months after catheter ablation, but only in males. Additionally, the type of coronary artery disease, namely ischemic heart disease due to obstructive atherosclerosis, was correlated with the recurrence of atrial fibrillation. Regarding the role of blood flow velocity measurement at the level of the LAA, it is known that a low velocity below 0.4 m/second is a predictor of the presence of thrombi at this level, while a higher value has the opposite effect, and at the same time, there are studies that have demonstrated that a higher velocity is associated both with higher chances of successful electrical cardioversion and with higher chances of maintaining the sinus rhythm for as long as possible [15].

Even if the studies in the specialized literature had as a standard a value of over 0.4 m/sec as a predictor of the success of the electrical cardioversion to sinus rhythm, but also of maintaining the sinus rhythm for a longer period of time, our study demonstrates that in order to maintain the rhythm sinus after ablation, speeds higher than 0.4 m/sec are needed, especially in the case of male patients. The average value of the speed measured in the group of men was 0.67 ± 0.26 m/sec, so a value above 0.4 m/sec and yet it was correlated with the risk of relapse at 3 months. Of course, to be able to establish the value of the speed at which the risk of relapse is low and the chances of maintaining the sinus rhythm for as long as possible, additional studies would be needed, which would include a significantly larger number of patients. In the present study, the type of coronary involvement was a predictor of relapse 3 months after catheter ablation. Ischemic heart disease due to macrovascular damage or obstructive atherosclerotic coronary disease was also a risk factor for the recurrence of atrial fibrillation after ablation.

Analyzing the questionnaires completed by the patients regarding the quality of life after catheter ablation, we observed a significant increase in the quality of life for both groups, but significantly more important in the female sex group. This result is extremely important, because it confirms the importance of performing the ablation procedure in women, who, as we well know, have a much worse quality of life and are more frequently symptomatic than men before undergoing interventional treatment [16]. An additional argument for the early interventional treatment in women is the fact that the female gender is not a risk factor for recurrence, at the same time they did not present significant complications during or after the procedure.

## 5. Conclusions

Women with atrial fibrillation and ischemic heart disease benefit much later from the catheter ablation procedure compared to men, and the causes responsible for this delay would be the more atypical symptoms, but also the greater number of antiarrhythmics used before the ablation procedure in women. The significant improvement in the quality of life, the absence of the female sex as a risk factor for relapse, but also the absence of gender differences regarding complications after ablation, are three very strong arguments to increase the addressability of women with atrial fibrillation to interventional treatment for heart rhythm control.

The limits of the study are mainly represented by the small number of included patients.

## Figures and Tables

**Figure 1 jcm-11-05568-f001:**
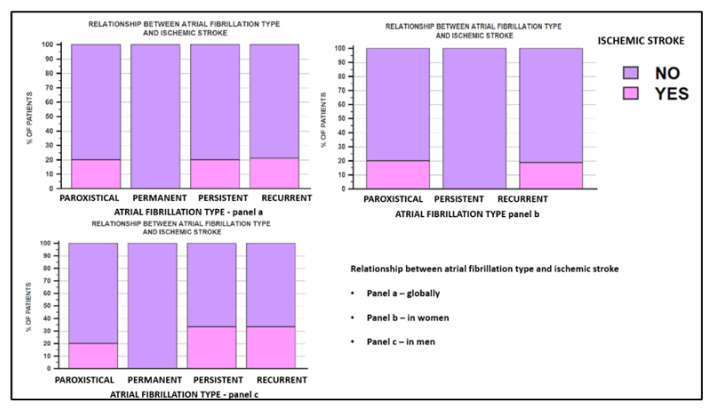
The relationship between the type of atrial fibrillation and ischemic stroke. In panel (**a**), the correlation between the 4 types of atrial fibrillation and the presence of ischemic stroke in the global study population is illustrated, and in panels (**b**,**c**) the correlation among female and male patients, respectively.

**Figure 2 jcm-11-05568-f002:**
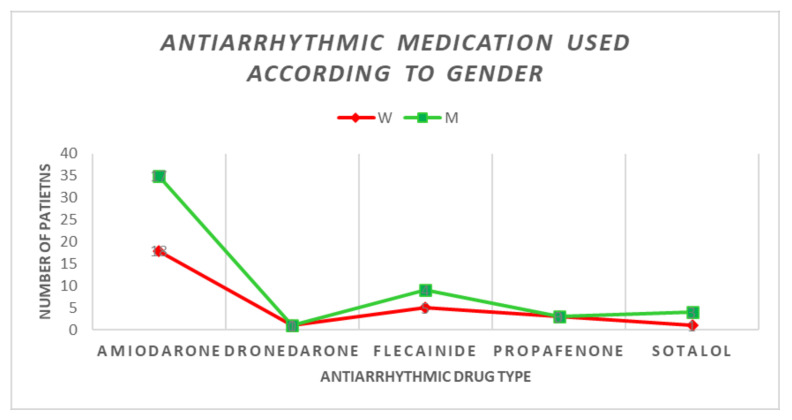
The main types of antiarrhythmics administered in the two groups.

**Figure 3 jcm-11-05568-f003:**
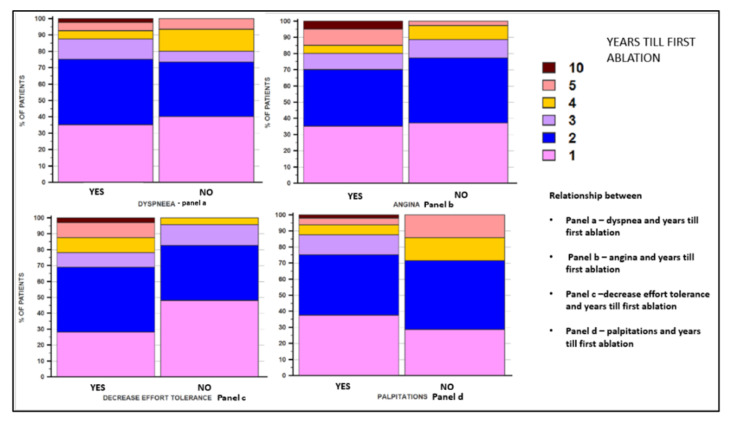
Correlation between patients’ symptoms and the number of years that passed until the ablation was performed, in the general population.

**Figure 4 jcm-11-05568-f004:**
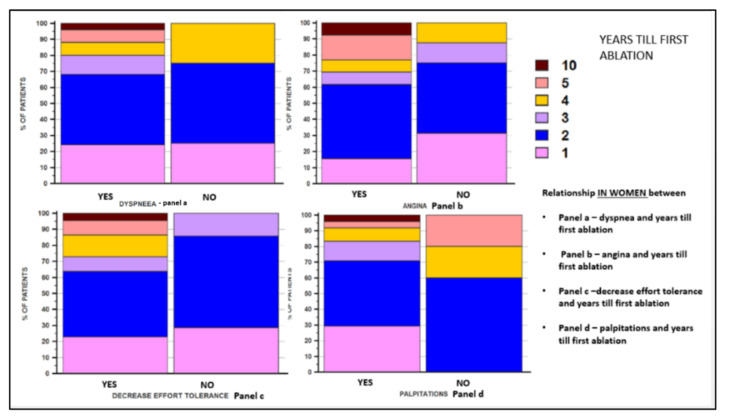
Correlation between the symptomatology of female patients and the number of years that passed until the ablation was performed.

**Figure 5 jcm-11-05568-f005:**
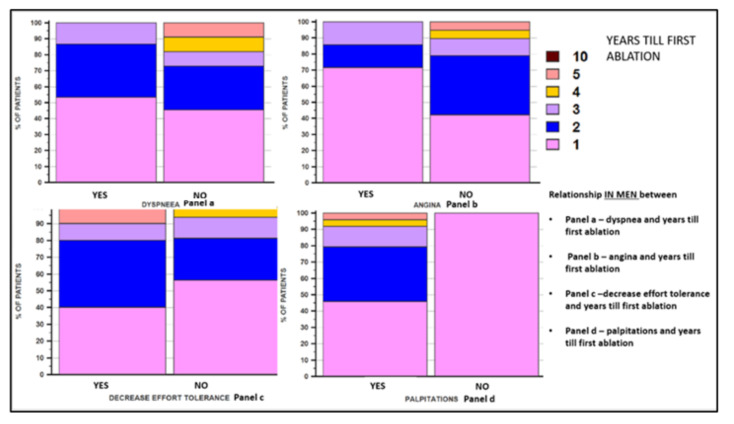
The correlation between the symptoms of male patients and the number of years that passed until the ablation was performed.

**Figure 6 jcm-11-05568-f006:**
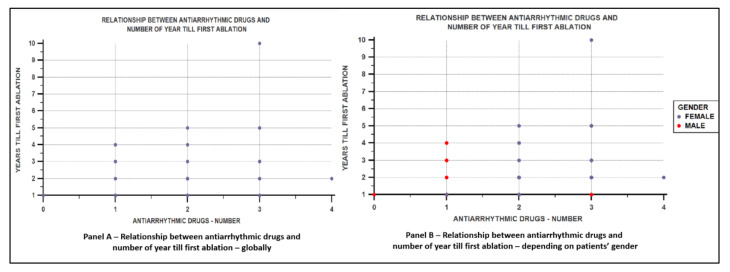
(**A**). Correlation between the number of antiarrhythmics used and the time elapsed until the ablation procedure was performed, estimated in years, in the global study population. (**B**). Correlation between the number of antiarrhythmics used and the time elapsed until the ablation procedure was performed, estimated in years, depending on the patient’s sex.

**Figure 7 jcm-11-05568-f007:**
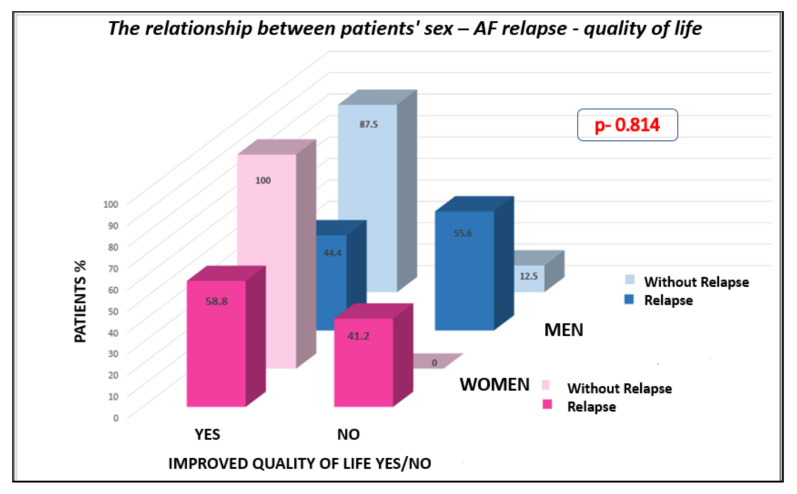
The relationship regarding the quality of life and the presence of relapse 3 months after the procedure, depending on the patient’s gender.

**Table 1 jcm-11-05568-t001:** The main characteristics of the patients included in the study.

Parameters		Female Sex	Male Sex	*p* Value
Number of patients		29 (52.7%)	26 (47.3%)	
Age	Mean ± SD	59.34 ± 8.99	61.69 ± 6.89	*p* = NS
Smoking	Yes	3 (10.34)	22 (84.61)	*p* < 0.0001
	No	26 (89.65)	4 (15.38)	
Alcohol	Yes	0 (0)	5 (19.23)	*p* = 0.044
	No	29 (100)	21 (80.76)	
Palpitation	Yes	24 (82.75)	24 (92.30)	*p* = NS
	No	5 (17.24)	2 (7.69)	
Decreased effort tolerance	Yes	22 (75.86)	10 (38.46)	*p* = 0.011
	No	7 (24.13)	16 (61.530	
Dyspnea	Yes	25 (86.20)	15 (57.69)	*p* = 0.038
	No	4 (13.79)	11 (42.30)	
Angina	Yes	13 (44.82)	7 (26.92)	*p* = NS
	No	16 (55.17)	19 (73.07)	
Mean HR *	Mean ± SD	77.34 ± 20.47	84.65 ± 22.40	*p* = NS
DPB *	Mean ± SD (Median)	80.17 ± 11.60 (80)	78.46 ± 10.07 (80)	*p* = NS
SBP *	Mean ± SD	135.17 ± 15.08	134.23 ± 17.18	*p* = NS
Total cholesterol	Mean ± SD	198 ± 59.48	173.73 ± 47.73	*p* = NS
LDL *	Mean ± SD (Median)	124.63 ± 54.08 (106)	104.57 ± 39.30 (100.5)	*p* = NS
HDL*	Mean ± SD	48.31 ± 11.10	42.69 ± 8.67	*p* = 0.043
Triglyceride	Mean ± SD	124.31 ± 52.19	154.30 ± 73.78	*p* = NS
Glucose level *	Mean ± SD (Median)	106.55 ± 25.51 (97)	106.53 ± 28.28 (99.5)	*p* = NS
Uric acid	Mean ± SD	5.88 ± 2.38	7.32 ± 1.9	*p* = 0.017
ESR *	Mean ± SD (Median)	16.20 ± 13.23 (12)	8.07 ± 4.84 (8)	*p* = 0.01
TSH *	Mean ± SD	2.89 ± 2.61	2.16 ± 0.97	*p* = NS
AF at admission	Yes	9 (31.03)	14 (53.84)	*p* = NS
	No	20 (68.96)	12 (46.15)	
	Paroxysmal	5 (17.24)	10 (38.46)	*p* = 0.006
	Persistent	8 (27.58)	12 (46.15)	
	Recurrent	16 (55.17)	3 (11.53)	
	Permanent	0 (0)	1 (3.84)	
Onset of AF (number of years) *	Mean ± SD (Median)	6.55 ± 2.65 (6)	5 ± 1.32 (5)	*p* = 0.014
Obesity	Yes	22 (75.86)	13 (50)	*p* = NS
	No	7 (24.13)	13 (50)	
Arterial Hypertension	Yes	28 (96.55)	24 (92.30)	*p* = NS
	No	1 (3.44)	2 (7.69)	
Microvascular IHD *	Yes	22 (75.86)	12 (46.15)	*p* = 0.047
	No	7 (24.13)	14 (53.84)	
Macrovascular IHD *	Yes	7 (24.13)	14 (53.84)	*p* = 0.047
	No	22 (75.86)	12 (46.15)	
Number of coronary lesions	One vessel	7 (100)	10 (71.42)	*p* = NS
	Two vessels	0 (0)	3 (21.42)	
	Multivessel	0 (0)	1 (7.14)	
Old MI *	Yes	1 (3.44)	7 (26.92)	*p* = 0.037
	No	28 (96.55)	19 (73.07)	
Diabetes mellitus	Yes	8 (27.58)	5 (19.23)	*p* = NS
	No	21 (72.41)	21 (80.76)	
HF *	Yes	11 (37.93)	7 (26.92)	*p* = NS
	No	18 (62.06)	19 (73.07)	
NYHA * CLASS	No HF	18 (62.06)	19 (73.07)	*p* = 0.046
	NYHA II	11 (37.93)	4 (15.38)	
	NYHA III	0 (0)	3 (11.53)	
	NYHA IV	-	-	
DCM *	Yes	1 (3.44)	4 (15.38)	*p* = NS
	No	28 (96.55)	22 (84.61)	
Thyroid pathology	Yes	14 (49.27)	9 (34.61)	*p* = NS
	No	15 (51.72)	17 (65.38)	
Other arrhythmias	Yes	16 (55.17)	13 (50)	*p* = NS
	No	13 (44.82)	13 (50)	
Ischemic stroke	Yes	4 (13.79)	7 (26.92)	*p* = NS
	No	25 (86.20)	19 (73.07)	

* Abbreviations meaning. HR—heart rate; DPB—diastolic blood pressure; SBP—systolic blood pressure; LDL—low-density lipoprotein; HDL—high-density lipoprotein; ESR—erythrocyte sedimentation rate; TSH—thyroid-stimulating hormone; AF—atrial fibrillation; IHD—ischemic heart disease; MI—myocardial infarction; HF—heart failure; NYHA—New York Heart Association; DCM—dilated cardiomyopathy.

**Table 2 jcm-11-05568-t002:** Echocardiographic and imaging parameters evaluated before performing atrial fibrillation catheter ablation.

Parameters		Global	Female Sex	Male Sex	*p* Value
Transthoracic echocardiography					
LA diameter	Mean ± SD (Median)	43.21 ± 6.10 (43)	41.75 ± 6.47 (41)	44.84 ± 5.31 (44)	***p* = 0.05**
LVEDD	Mean ± SD (Median)	50.36 ± 6.21 (50)	48.13 ± 4.8 (46)	52.84 ± 6.73 (50)	***p* = 0.005**
LVESD	Mean ± SD (Median)	37.7 ± 27.26 (34)	39.62 ± 36.96 (34)	35.61 ± 8.16 (35)	*p* = 0.288
LVEF%	Mean ± SD (Median)	50.34 ± 5.99 (50)	51.51 ± 5.34 (50)	49.03 ± 6.50 (50)	*p* = 0.174
Diastolic dysfunction	Yes	51 (92.7%)	27 (93.10%)	25 (96.15%)	*p* = 0.148
	No	4 (7.3%)	2 (6.9%)	1 (3.85%)	
	Grade I	45 (88.2%)			*p* = 0.548
	Grade II	5 (9.8%)			
	Grade III	1 (2%)			
Mitral insufficiency	Yes	52 (94.5%)	27 (93.10%)	25 (96.15%)	*p* = 0.922
	No	3 (5.5%)	2 (6.9%)	1 (3.85%)	
	Mild	45 (88.2%)	23 (92%)	22 (84.61%)	*p* = 0.548
	Moderate	5 (9.8%)	2 (8%)	3 (11.53%)	
	Severe	1 (2%)	0 (0%)	1 (3.84%)	
Pulmonary hypertension	Yes	20 (36.4%)	11 (37.9%)	9 (34.61%)	*p* = 0.979
	No	35 (63.6%)	18 (62.1%)	17 (65.4%)	
Transesophageal echocardiography					
LAA diameter	Mean ± SD (Median)	1.75 ± 0.47 (1.8)	1.45 ± 0.34 (1.4)	2.08 ± 0.37 (2.1)	*p* < 0.0001
LAA velocity	Mean ± SD (Median)	0.66 ± 0.23 (0.66)	0.59 ± 0.16 (0.6)	0.74 ± 0.28 (0.75)	*p* = 0.0263
Spontaneous echo contrast LAA	Yes	8 (14.5 %)	3 (10.33%)	5 (19.23%)	*p* = 0.5822
	No	47 (85.5%)	26 (89.65%)	21 (80.76%)	
LAA thrombus in the past	Yes	3 (5.5%)	1 (3.44%)	2 (7.69%)	*p* = 0.922
	No	52 (94.5%)	28 (96.56%)	24 (92.31%)	
PFO	Yes	9 (16.4%)	8 (27.58%)	1 (3.89%)	*p* = 0.044
	No	46 (83.6%)	21 (72.41%)	25 (96.15%)	
Cardiac and pulmonary vein CT angiogram					
Indication for CT		54 (98.2%)	28 (96.55%)	26 (100%)	*p* = 0.956
4 PV	Yes	54 (98.2%)	28 (96.55%)	26 (100%)	*p* = 0.960
	No	1 (1.8%)	1 (3.45%)	0 (0%)	
PV anomaly	Yes	15 (27.8%)	10 (38.46%)	5 (19.23%)	*p* = 0.378
	No	40 (72.2%)	19 (61.53%)	21 (80.77%)	
	APVD	1 (7.1%)	1 (3.44%)	0 (0%)	*p* = 0.956
	Common left-sided trunk	13 (92.9%)	10 (34.4%)	3 (11.53%)	*p* = 0.045

LA—left atrium; LVEDD—left ventricle end-diastolic diameter; LVESD—left ventricle end-systolic diameter; LVEF%—left ventricle ejection fraction; LAA—left atrial appendage; PFO—patent foramen ovale; CT—computer tomography; PV—pulmonary vein; APVD—anomalous pulmonary venous drainage.

**Table 3 jcm-11-05568-t003:** Aspects related to the treatment used to prevent thromboembolic events.

		Female Sex	Male Sex	*p* Value
CHA2DS2-VASc Score	Median	3	3	*p* = 0.166
HAS-BLED Score	Median	1	1	*p* = 0.946
Anticoagulant	Yes	89.66% (26/29)	100% (26/26)	*p* = 0.274
	No	10.34% (3/29)	0% (0/26)	
	DOACs	46.15% (12/26)	57.7% (15/26)	*p* = 0.578
	VKA	53.85% (14/26)	42.3% (11/26)	
Antiplatelet medication	Yes	6.9 % (2/29)	15.38 % (4/26)	*p* = 0.565
	No	93.1% (27/29)	84.62% (22/26)	
	Aspirin	50% (1/2)	50% (2/4)	*p* = 0.386
	Clopidogrel	50% (1/2)	50% (2/4)	

DOACs—direct oral anticoagulants; VKA—vitamin K antagonist.

**Table 4 jcm-11-05568-t004:** The main therapeutic classes administered for heart rate control, both in the general population and according to the patient’s sex.

Drug Class		Global	Female Sex	Male Sex	*p* Value
Beta-blockers		46/55 (83.63%)	25/29 (86.2%)	21/26 (80.7%)	*p* = 0.898
Beta-blocker type	Bisoprolol	15 (32.6%)	5 (33.3%)	10 (66.66%)	*p* = 0.05
	Metoprolol	26 (56.52%)	19 (73.07%)	7 (26.93%)	
	Propranolol	2 (4.34%)	1 (50%)	1 (50%)	
	Nebivolol	2 (4.34%)	0 (0%)	2 (100%)	
	Carvedilol	1 (2.17%)	1 (100%)	0 (0%)	
Non-dihydropyridinic CCBs		0 (0%)	0 (0%)	0 (0%)	*p* = 0.979
Digoxin		5 (9.1%)	3 (60%)	2 (40%)	*p* = 0.922

CCBs—calcium-channel blockers.

**Table 5 jcm-11-05568-t005:** Treatment aspects for cardiac rhythm control: electrical cardioversion and catheter ablation.

		Global	Female Sex	Male Sex	*p* Value
Pharmacological cardioversion	Yes	22 (40.0%)	13 (44.82)	9 (34.61)	*p* = 0.619
	No	33 (60.0%)	16 (55.17)	17 (65.38)	
Electrical cardioversion (EC)	Yes	23 (41.8%)	8 (27.58)	15 (57.69)	*p* = 0.047
	No	32 (58.2%)	21 (72.41)	11 (42.30)	
Number of EC *	Median (MIN–MAX)	0 (0–4)	0 (0–4)	1 (0–3)	*p* = 0.113
Years until the first ablation *	Mean ± SD (Median)	2.2 ± 1.55 (2)	2.55 ± 1.84 (2)	1.80 ± 1.05 (1.50)	*p* = 0.05
Indication for ablation	Paroxysmal	15 (27.3%)	5 (17.24)	10 (38.46)	*p* = 0.006
	Recurrent	19 (34.5%)	16 (55.17)	3 (11.53)	
	Persistent	20 (36.4%)	8 (27.58)	12 (46.15)	
	Permanent	1 (1.8%)	0 (0)	1 (3.84)	
Type of ablation	Radiofrequency	49 (89.1%)	26 (89.65)	23 (88.46)	*p* = 0.770
	Cryoablation	6 (10.9%)	3 (10.34)	3 (11.53)	
EC during ablation	Yes	21 (38.2%)	11 (38.93)	10 (38.46)	*p* = 0.812
	No	34 (61.8%)	18 (62.06)	16 (61.53)	
Number of electrical shocks during the ablation *	0	34 (61.8%)	18 (62.06)	16 (61.53)	*p* = 0.212
	1	17 (30.9%)	7 (24.13)	10 (38.46)	
	2	2 (3.6%)	2 (6.89)	0 (0)	
	3	2 (3.6%)	2 (6.89)	0 (0)	
Complication	Yes	5 (9.1%)	2 (6.9%)	3 (11.54%)	*p* = 0.328
	No	50 (90.9%)	27 (93.1%)	23 (88.46%)	
	Femoral hematoma	2 (40%)	0 (0%)	2 (7.69%)	
	Small pericardial effusion	2 (40%)	2 (6.89%)	0 (0%)	
	Tamponade	1 (20%)	0 (0%)	1 (3.84%)	

* It does not respect the conditions of normality. EC—electrical cardioversion.

**Table 6 jcm-11-05568-t006:** Risk factors for recurrence of atrial fibrillation 3 months after catheter ablation.

			Relapse of AF	Women		Men		Global	
Sex			Yes	17 (58.6%)	-	18 (69.2%)	-	35 (63.6%)	
			No	12 (41.4%)		8 (30.8%)		20 (36.4%)	
Age			Yes	57.47 ± 10.39	*p* = 0.186	62.22 ± 5.47	*p* = 0.650	59.91 ± 8.46	*p* = 0.517
			No	62 ± 5.98		60.50 ± 9.73		61.40 ± 7.50	
LDL-cholesterol			Yes	124.76 ± 48.46	*p* = 0.984	97.72 ± 29.39	*p* = 0.187	110.85 ± 41.53	*p* = 0.371
			No	125.16 ± 63.47		120 ± 55.02		123.10 ± 58.77	
Uric Acid			Yes	6.32 ± 2.75	*p* = 0.244	6.92 ± 1.60	*p* = 0.104	6.63 ± 2.22	*p* = 0.781
			No	5.26 ± 1.64		8.23 ± 2.30		6.45 ± 2.39	
LAA velocity			Yes	0.58 ± 0.15	*p* = 0.738	0.67 ± 0.26	*p* = 0.05	0.62 ± 0.22	*p* = 0.160
			No	0.60 ± 0.17		0.89 ± 0.27		0.72 ± 0.25	
AF type	Paroxysmal		Yes	2 (11.76%)	*p* = 0.441	5 (27.77%)	*p* = 0.295	7 (20%)	*p* = 0.35
			No	3 (25%)		5 (62.5%)		8 (40%)	
	Recurrent		Yes	11 (64.7%)		3 (16.66%)		14 (40%)	
			No	5 (41.66%)		0 (0%)		5 5(25%)	
	Persistent		Yes	4 (23.52%)		9 (50%)		13 (37.14%)	
			No	4 (33.33%)		3 (37.5)		7 (35)	
	Permanent		Yes	0 (0%)		1 (5.55%)		1 (2.85%)	
			No	0 (0%)		0 (0%)		0 (0%)	
IHD type	Microvascular		Yes	50%	*p* = 0.218	58.33%	*p* = 0.491	52.94%	*p* = 0.007
			No	50%		41.67%		47.1%	
	Macrovascular		Yes	85.71%		78.58%		80.95%	
			No	14.29%		21.42%		19.05%	
Thyroid pathology		With	Yes	10 (58.82%)	*p* = 0.329	8 (44.44%)	*p* = 0.256	18 (51.42%)	*p* = 0.103
			No	4 (33.33%)		1 (12.5%)		5 (25%)	
		Without	Yes	7 (41.17%)		10 (55.55%)		17 (48.57%)	
			No	8 (66.6%)		7 (87.5%)		15 (75%)	

AF—atrial fibrillation; LDL—low-density lipoprotein; LAA—left atrial appendage; IHD—ischemic heart disease.

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
