# Peer review of "Particularities of Catheter Ablation in Women with Atrial Fibrillation and Associated Ischemic Heart Disease"

_jcm, 2022, doi:10.3390/jcm11195568_

Round 1

Reviewer 1 Report

The manuscript "Particularities of catheter ablation in women with atrial fibrillation and associated ischemic heart disease" by Irimie D et al. describes the particularities of ablation in women with atrial fibrillation and ischemic heart disease. There are some comments.

Page 3, Table 1.

Comment 1: Please provide information regarding values of BNP or NT pro BNP if possible. A high BNP or NT pro BNP value indicates a latent heart failure that would influence the choice of beta-blockers and anti-arrhythmic drugs. 

Page 8, line 198-200

"An alternative to the administration of beta-blockers is the administration of non-dihydropyridine calcium channel blockers, but these are much less frequently used compared to beta-blockers."

Comment 2: Table 4 shows that no patient administered non-dihydropyridine calcium channel blockers as an alternative to beta-blockers. Therefore, this sentence's description, "these are much less frequently" would be inappropriate.

Page 9, Table 4 first row

Comment 3: The female sex column for Bisoprolol lacks parenthesis for percentage.

Page 9, Line 210

"As I said before"

Comment 4: "I "would be "we." 

Page 16, Line 374

"satisfactory, These"

Comment 5: The comma would be a period.

Page 17, Line 419

"it is known that a low velocity below 4 cm/second"

Comment 6: "4 cm/second" would be "40 cm/second" or "0.4 m/second". It might be better to unify the velocity "40 cm/second" to "0.4 m/sec," as stated in lines 424 and 427.

Page 17, line 452

"but also the absence of complications after the procedure"

Comment 7: Table 5 shows the complications in the Female sex as for the Male sex. This sentence would be inappropriate. Moreover, the conclusions appear inconsistent between the abstract and the text.

Author Response

First of all, I would like to thank you for your time, but especially for the very relevant comments, which really add value to this work. I will respond punctually for each individual comment. I also took into account everything you suggested, thus making changes in the manuscript as well.

Page 3, Table 1.

Comment 1: Please provide information regarding values of BNP or NT pro BNP if possible. A high BNP or NT pro BNP value indicates a latent heart failure that would influence the choice of beta-blockers and anti-arrhythmic drugs. 

Response 1: Indeed, the value of NT pro BNP or BNP would have helped us a lot for the diagnosis of heart failure. In the present case, the diagnosis for HF was established based on the symptoms and the physica exam. For objective reasons we did not manage to add the value of these biomarkers. A large part of the patients included in the study did not have these biomarkers dosed (the reason being the lack of kits), so due to the small number of doses, we decided not to include the value of NT-pro BNP and BNP in our work.

Page 8, line 198-200

"An alternative to the administration of beta-blockers is the administration of non-dihydropyridine calcium channel blockers, but these are much less frequently used compared to beta-blockers."

Comment 2: Table 4 shows that no patient administered non-dihydropyridine calcium channel blockers as an alternative to beta-blockers. Therefore, this sentence's description, "these are much less frequently" would be inappropriate.

Response 2: Indeed, you are right, Thank you for this observation.

Page 9, Table 4 first row

Comment 3: The female sex column for Bisoprolol lacks parenthesis for percentage.

Response 3: Thank you, I added the percentage. it was a small inattention.

Page 9, Line 210

"As I said before"

Comment 4: "I "would be "we." 

Response 4: Thank you, you are right.

Page 16, Line 374

"satisfactory, These"

Comment 5: The comma would be a period.

Response 5: Indeed. The correction was made.

Page 17, Line 419

"it is known that a low velocity below 4 cm/second"

Comment 6: "4 cm/second" would be "40 cm/second" or "0.4 m/second". It might be better to unify the velocity "40 cm/second" to "0.4 m/sec," as stated in lines 424 and 427.

Response 6: That was by mistake. I corrected and unify to be 0.4 m/sec everywhere in the text.

Page 17, line 452

"but also the absence of complications after the procedure"

Comment 7: Table 5 shows the complications in the Female sex as for the Male sex. This sentence would be inappropriate. Moreover, the conclusions appear inconsistent between the abstract and the text.

Response 7: Indeed, the expression is not appropriate and correct, but I reformulated that sentence. Practically, we wanted to emphasize that although the presence of post-procedural complications was more often incriminated in women as a reason for performing the ablation later than in men, through our study we demonstrated that there are no differences between the two sexes. And, as for the part of the differences between the abstract and the text, for objective reasons (the number of words is limited) we tried to include a conclusion as short as possible in the abstract.

Reviewer 2 Report

This is an interesting study with the clinical usefulness and implication. However, there are some minor comments.

1. In the table2, the percentage of pulmonary hypertension is considerably high.

Please clarify the reason for this phenomenon and how to evaluate pulmonary hypertension by TTE.

2. P17 L 414-416 "Among them, the velocity of the blood flow from the LAA was found to be associated with the risk of relapse 3 months after catheter ablation, but only in males.”

I would like to show me the basis on which you arrived at this consideration.

Author Response

First of all, I would like to thank you for your time and I am glad that you found this work interesting. I would also like to thank you for the extremely pertinent comments that you made. I will respond promptly to each individual comment.

Comment 1. In the table2, the percentage of pulmonary hypertension is considerably high.

Please clarify the reason for this phenomenon and how to evaluate pulmonary hypertension by TTE.

Response 1: Indeed, a rather large percentage of patients presented pulmonary arterial hypertension, and the explanation underlying this phenomenon is the multiple cardiac pathology affecting the left heart, such as diastolic and systolic dysfunction, the presence of mitral valve pathology. Also, a large part of the patients were also smokers, therefore, it is possible that a respiratory component not diagnosed until that moment contributed to the appearance of pulmonary hypertension. In TTE, pulmonary hypertension was evaluated based on the pressure gradient in the right cavities, the transtricuspid flow velocity, and the pressure in the right atrium. The value of the systolic arterial pressure in the pulmonary artery was practically calculated. A value greater than 30 mmHg being considered abnormal.

Comment 2. P17 L 414-416 "Among them, the velocity of the blood flow from the LAA was found to be associated with the risk of relapse 3 months after catheter ablation, but only in males.”

I would like to show me the basis on which you arrived at this consideration.

Response 2: In table 6 we presented the risk factors/parameters that correlated with the recurrence of atrial fibrillation 3 months after ablation. In addition to the patients' sex, age, type of atrial fibrillation, form of ischemic heart disease, the level of inflammation evaluated by the value of uric acid, the presence of thyroid pathology, we also evaluated the value of the blood flow velocity in the LAA. In the statistical analysis, the velocity value was associated with the recurrence of atrial fibrillation, but only among male patients.
